# Unsupervised Cognition

## Abstract

Unsupervised learning methods have a soft inspiration in cognition models. To this day, the most successful unsupervised learning methods revolve around clustering samples in a mathematical space. In this paper we propose a primitive-based unsupervised learning approach inspired by novel cognition models. This representation-centric approach models the input space constructively as a distributed hierarchical structure in an input-agnostic way. We compared our approach with the current state-of-the-art in unsupervised learning classification. We show how our proposal performs better than any of the alternatives. We also evaluate some cognition-like properties of our proposal that other algorithms lack, even supervised learning ones.

## 1 Introduction

Unsupervised learning is a huge field focused on extracting patterns from data without knowing the actual classes present in the data. Therefore, the field is full of methods that cluster data from the properties of their mathematical representation. This hampers their applicability to data whose mathematical relationships do not directly correlate with their cognitive relationships, that are those that cognitive agents (like humans) find between the data. For example, the MNIST dataset has a clear cognitive relationship between its elements: the number they represent. However, when transformed into numerical values for clustering, their relationships fade out in favour of relationships between their encodings, that do not necessarily correspond with the cognitive ones.

In this field, the most well-known and validated algorithm is the K-Means algorithm. It is a clustering algorithm that decides if two inputs belong to the same class based on the euclidean distance between their encodings. Regarding grey image clustering, the state-of-the-art algorithm is the Invariant Information Clustering (IIC) (Ji et al., 2019) algorithm, whose goal is to extract common patterns directly from the images using convolutions and neural networks. What is common to both algorithms (as well as most of the algorithms from the field) is that they assume there is an infinite mathematical space, and their learning focus on shaping the limits between classes in that space. In that regard, their representations are spatial, in the sense that the classes are represented by subspaces of an infinite space. Thus, if an input falls into a subspace, it is classified with the corresponding label of such subspace, independently of its relationship to the rest of the samples of the subspace. This in fact allows to classify samples that are not similar to anything seen during learning, although those classifications are not always very accurate.

In contrast, novel theories about how the brain works propose that the brain models the world in a constructive way, that is, it generates constructive representations of the world. It would do so to be able to predict the world (Hawkins & Blakeslee, 2004; Leibo et al., 2015; Yon et al., 2020). A constructive representation would be an abstraction or archetype of a class, in the sense that it would be a representation to which any (or at least some) elements of the class are similar to. This implies that, to assign a class to an input, it has to be similar enough to one of the already learned representations, and if it is not similar enough to any of them then it can not be classified. Mathematically speaking, the difference between both approaches is that the first, traditional one focus on splitting a representation space, and the second, novel one focus on building a representations set.

To empirically evaluate the performance of this new approach, we propose a novel unsupervised learning algorithm whose goal is to model the input space generating constructive representations. This implies that, in a way, our proposal has to be able to automatically generate abstractions from the inputs. To do so, it requires a representation-oriented universal data structure. Recent research

has shown that such data structure is a Sparse Distributed Representation (Ahmad & Hawkins, 2016; Cui et al., 2017) (SDR), which allows an universal representation of the inputs independently of their type. This has been proven to be the actual way in which the brain processes its inputs (Foldiak, 2003; Cui et al., 2017). Thus, we will need a method to transform our input spaces into SDRs. We will do so through an encoder decoder pair we call *Embodiment*.

Using SDRs as inputs, we propose an unsupervised learning algorithm that models the input space generating constructive representations in an input-agnostic way. We call such algorithm a *Primitive* and we expect it to be the building block of a future cognition algorithm. In this paper we present its essence as well as one of its modulators: the *Spatial Attention* modulator. This modulator will auto-regulate the spatial discriminability of the algorithm. Additionally, we developed our proposal to be transparent and explainable, as it is desirable that any solution can describe its representations and explain its decisions. Finally, a perk of focusing on generating constructive representations is that our algorithm is able to state if a new input does not correspond to any previously seen pattern.

We compared our proposal to the main unsupervised classification algorithms: K-Means for tabular data and IIC for grey image data. We compared it with different configurations of K-Means and IIC and for four different static datasets (two tabular and two image datasets). The goal was a classification task in which K-Means labels were assigned after training based on the most common label present in each cluster. The results show a clear advantage of our proposal, being able to deal with both tabular and image data with a decent performance. Additionally, we performed some experiments to evaluate cognition-like properties. In this case we compared our proposal to not only K-Means and IIC, but also other clustering supervised method, K-NN. The comparison consisted on recognising MNIST digits even when removing random pixels. In that experiment there is a clear advantage of our proposal over the compared algorithms that shows how building constructive representations produce a different behaviour. We consider that these results show that algorithms focused on building constructive representations have the potential to have cognition-like properties. Thus, we conclude that our proposal is a disruptive unsupervised learning algorithm, with different, more promising properties than traditional algorithms.

The rest of the paper is organised in a related work resume at Section 2, a proposal description in Section 3, an empirical evaluation at Section 4, a discussion in Section 5, and a resume of the conclusions and future work at Section 6.

## 2 RELATED WORK

There are multiple algorithms for unsupervised learning developed along the years, from generic clustering algorithms like K-Means (Lloyd, 1982), to more specific, usually neural network based, algorithms that deal with only one task. In this second category we can find algorithms that deal with representation learning (Wang et al., 2020), video segmentation (Araslanov et al., 2021) or speech recognition (Baevski et al., 2021). However, none of them try to build constructive representations, but instead they divide a mathematical representation of the input space into clusters that are supposed to represent the different classes present in it.

Among these clustering algorithms, there are few that stand out, specially for the task of unsupervised classification. One of them is K-Means due to its performance clustering tabular data. This algorithm clusters the samples based on their closeness in the mathematical space of their encodings. Another one is Invariant Information Clustering (IIC) (Ji et al., 2019) due to its performance clustering grey images. This algorithm takes an image, transforms it with a given transform, and then run both of them over two neural networks with the goal of learning what is common between them. To that effect, it looks to maximise the mutual information between encoded variables, what makes representations of paired samples the same, but not through minimising representation distance like it is done in K-Means. In any case, both algorithms stand out due to their performance in their respective domains, but none of them is able to obtain good accuracy across domains. Thus, we will use them as baseline for comparison purposes, even though they cannot be applied in all cases.

Finally, regarding brain-inspired methods that try to predict the input space, the only research we are aware of is the Hierarchical Temporal Memory (Cui et al., 2017) (HTM) and SyncMap (Vargas & Asabuki, 2021), although they are algorithms suited for learning sequences instead of static data, and

HTM is not unsupervised. Thus, as far as we are aware, ours is the first proposal of a brain-inspired constructive unsupervised learning algorithm for predicting static data.

## 3    THE PROPOSAL

Our proposal is composed by an Embodiment, a Primitive and a Spatial Attention modulator. The goal of the Embodiment is to transform the input space into Spatial Distributed Representations (SDRs), the goal of the Primitive is to process those SDRs and model the input space generating constructive representations, and the goal of the Spatial Attention modulator is to help the Primitive to self-regulate.

### 3.1    THE EMBODIMENT

To translate inputs to SDRs we need an encoder architecture. To interpret the SDRs the Primitive generates we need a decoder architecture too. Both architectures conform the Embodiment of our Primitive. In our case, as in our experiments we only explore tabular and grey image datasets, we only present three kinds of encoder decoder pairs: one for float point numbers, one for categorical data, and one for grey images.

The grey images translation to SDR is straightforward: a grey image's SDR is a flattened version of the image (where each pixel is a dimension) with the values normalise to be between $0$ and $1$. In the case of float point numbers, its translation to SDR is a bit more nuanced. We take the input space of the number (that is, the possible values it can take) and divide it into bins. Those bins will be the dimensions of the SDR. Then, each number will fall into one of those bins. However, in order to allow some overlap between numbers (what is fundamental for finding similarities using our Primitive), we also activate as many bins around the number bin as another parameter we call *bin overlap*. With this, the SDR is a long list of zeros and some ones around the bin where the number falls. By default, we use an overlap of $10\%$ of the number of bins, that by default is set to $100$ bins. In the case of categorical data we create one bin per category and set the overlap to $0\%$.

Having these representations, we define the SDR representation of a tabular input as a concatenation of the SDR representations of each entry, adjusting the indices to put one entry representation after another. Using this same methodology, we can compose multiple input types into one SDR. Although this is not the goal of this paper, these compositions could potentially help our algorithm to deal with datasets where the input has many different types (for example, keyboard keys and video, like the recently released MineRL dataset (Guss et al., 2019)), although proving that would be matter of future work.

Here it is important to remark that, due to the transformation of any input and any output into SDRs, the algorithm always deals with the same data structures, and thus it is optimised to learn them independently of what they represent. Moreover, this makes our algorithm input-agnostic, as any kind of data is potentially transformable into an SDR.

### 3.2    THE PRIMITIVE

Once we have an SDR representation of the inputs (and a way to recover the values from the SDR representation), we need to process them. To that end we need an internal representation of the SDRs, that we called a *Footprint*. This Footprint contains an SDR, and can contain (for evaluation purposes only) a label value. This Footprint also has an updating function and an activation function. The updating function modifies the SDR when necessary mixing the Footprint SDR with an external SDR, while the activation function computes an SDR containing a weighted version of the Footprint SDR. The Footprint updating and activation functions are displayed in Algorithms 1 and 2 respectively. To compute the similarity score that appears in the activation function we need a *Similarity Function* that compares SDRs. This Similarity Function should take two SDRs and return a similarity value stating how similar we consider them to be. The specific Similarity Function we developed for this paper is a variation of the euclidean distance, but we also tested using the euclidean distance between vectors and the differences in results are minimal.

As it is clear from Algorithm 1, our approach to build constructive representations consists on merging similar inputs to build the abstraction or archetype. The core of our algorithm revolves around

**Algorithm 1** Footprint Update (all ops are element-wise)

**Require:** $FP$: a Footprint, $In$: an input
1: $fp \leftarrow FP.SDR \quad inp \leftarrow In.SDR$
2: $n \leftarrow FP.N$ {Recall #inputs mixed into the Footprint}
3: $tmp \leftarrow fp * n$
4: $tmp \leftarrow tmp + inp$
5: $FP.SDR \leftarrow tmp/(n + 1)$
6: $FP.N \leftarrow n + 1$

**Algorithm 2** Footprint Activation (all ops are element-wise)

**Require:** $FP$: a Footprint, $In$: an input
1: $fp \leftarrow FP.SDR \quad inp \leftarrow In.SDR$
2: $FP.ACTIVATION \leftarrow fp * similarity(fp, inp)$
3: **return** $FP.ACTIVATION$

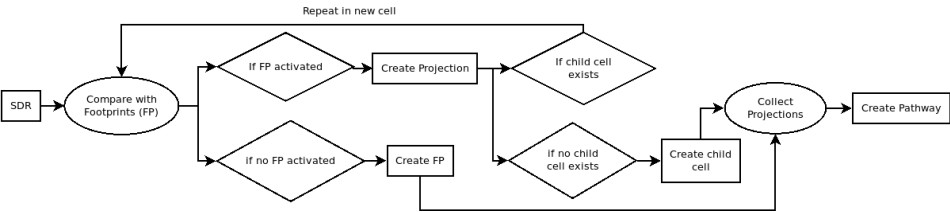

Figure 1: Primitive Training Schema

building and organising these representations in the form of Footprints. To organise them, they are grouped inside the nodes of a tree structure. These nodes (that we call cells) contain a set of Footprints and a threshold common to all their respective Footprints. This threshold will be defined in deep in the next section. At the beginning of training the only node is the root node (that we call seed cell), that starts with no Footprints. When new inputs are processed during training, new Footprints are generated and the seed cell grows. And when an input is considered similar to an existing Footprint, then a new cell is created that is a child of the seed cell, and is associated with that Footprint. Thus, a cell can have as many children as Footprints, and each child cell has an associated parent Footprint. To decide if a new input is considered similar to an existing Footprint, we use our Similarity Function to get an score of the similarity of both SDRs, and if that score is over the cell threshold, then both the input and the Footprint are considered similar to each other. When there are more than one Footprint with a similarity over the cell threshold, we consider similar only the one with higher similarity. This similar Footprint is then the *active* Footprint.

Now let us show how a new input is processed by out Primitive. To follow this description, a general schema of this algorithm is displayed at Figure 1. We start the process with an SDR that is a new input. This input is then provided to the seed cell to be compared against the existing Footprints. If any Footprint is activated as explained before, then the updating and activation functions are executed, in that order, over the activated Footprint. The output of the activation function is what we call a *Projection*, and it becomes the output of the cell. If no Footprint is activated, that is, the input is considered different to all the Footprints of the cell (or the cell is empty), then there is no Projection and, if we are in training mode, a new Footprint is created.

Whenever a Footprint is activated by multiple inputs, a new child cell is generated that is associated to such Footprint. To generate such new cell we require that at least two inputs activated that Footprint, that is, they had their highest similarity with such Footprint and such similarity was over the cell threshold. If an activated Footprint has an associated child cell, the input is passed down to that cell to repeat the process there and an additional Projection, and thus a cell output, is generated. If an activated Footprint does not have an associated child cell, or none of the Footprints of the current cell are activated, then the downward pass ends.

Our Primitive also has an upward pass, whose goal is to increase the robustness of the Footprints. This upward pass takes the last cell with an output and passes its output to its parent cell. This output is then processed in the parent cell, that is, the parent cell process the output of its child cell, and consequently updates its output. This behaviour is repeated again until the seed cell has processed the output of its child cell. Processing a child's output modifies the actual values of the activated

**Algorithm 3** Spatial Attention

**Require:** $CELL$: a cell
**Require:** $INPUT$: a new input
1: $input_{last} \leftarrow CELL.LastInput.SDR$
2: $input_{new} \leftarrow INPUT.SDR$
3: $SA \leftarrow similarity(input_{last}, input_{new})$

4: **return** $SA$

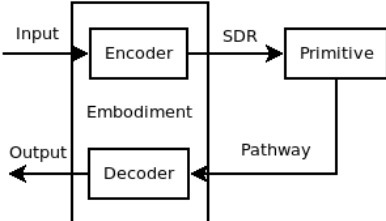

Figure 2: Global Schema

Footprint SDR, diluting the fine detail the Footprint has learned from the input, thus giving more robustness to it. The upward pass is not represented in Figure 1 to keep it simple and readable, but it follows the same schema, only changing the input SDR by the child projection, and the conditions on child cell by conditions on parent cell.

Finally, the average of the outputs of each cell is the output of the Primitive, what we call a *Pathway*. This Pathway is an instance of the internal representation of the given input, and thus it will be similar to it, but at the same time it will contain the information about the label associated with the internal representation, thus providing a classification for the input. This Pathway is later processed by the decoder of the Embodiment to retrieve the final label of the input. A resume of the global algorithm is displayed at Figure 2. It is important to note that, if we are in evaluation mode, then the update function of the Footprints is not executed to not modify the internal representations, and no new Footprints neither cells are created.

To end with the Primitive, it is important to note that each cell has its own similarity threshold. This allows for a better discrimination policy, as we will see in the next section. It is also important to note that the existence of a threshold also in the seed cell allows to determine if an input is recognised or not by the algorithm, as any input will produce an "I do not know" answer if it does not produce a similarity over the threshold with any Footprint from the seed cell. Additionally, it is important to note that, during training the label is provided to be able to include it in the Footprints, but it is not used for comparing Footprints, and thus for building and organising the internal representations. Later, when in evaluation mode, the label of each Footprint would be a mixture of the labels of each input that activated such Footprint, and when classifying the returned label will be the strongest label of the Pathway. Finally, it is important to note how our Primitive is representation-centric, as the whole algorithm focuses on generating the right internal representations of the inputs, what would produce the right Pathways for classification.

### 3.3 THE SPATIAL ATTENTION MODULATOR

In the previous Section, a threshold was used to decide if two SDRs where considered similar or not. This threshold can be set arbitrarily, but that would hamper the performance of the Primitive and would generate multiple extra parameters of the model. Thus, a way to automatise the threshold selection was needed, and that is the role of the Spatial Attention Modulator. The whole role of this modulator is to measure the variability of the input space of the cell and dynamically set a similarity threshold. The algorithm we developed to set such threshold is the similarity between the new input and the last input, and its pseudo-code is displayed in Algorithm 3.

The rationale behind using this approach is that any input space has a certain variability, and the right threshold will be that one that sits in the middle of such variability. Assuming the dataset has been randomly shuffled, this variability can be approximated by the similarity between two consecutive inputs. This actually allows for child cells to have higher thresholds than their parent cells, as their input space are limited to those inputs that are similar to their associated parent Footprint. This generates an increase in discrimination power the further down the cell hierarchy a Footprint is. In turn, this develops a distributed hierarchy, where each cell process a different subdomain of the input domain.

| Name | Type | # Features | # Samples |
|------|------|-----------|-----------|
| Wisconsin Breast Cancer | Tabular (Numerical) | 30 | 569 |
| Pima Indians Diabetes | Tabular (Numerical) | 8 | 768 |
| MNIST | Image (B&W) | $28 \times 28$ | $60,000 + 10,000$ |
| ImageNet-1k | Image (B&W) | $256 \times 256$ | $1,281,167 + 50,000$ |

Table 1: Characteristics of the experimental subjects

## 4 Empirical Evaluation

To evaluate our proposal, we performed two different experiments: a comparison in classification task versus other unsupervised learning algorithms, and a comparison in cognition-like capabilities versus other clustering algorithms. All the experiments were run in an Ubuntu laptop with an Intel Core i9-13900HX at 2.60GHz with 32 cores, 32Gb of memory, and a NVIDIA GeForce RTX 4060 with 8Gb of VRAM.

### 4.1 Experimental Subjects

Our experimental subjects for these experiments where four datasets: two tabular datasets full of numerical values, and two image datasets of grey images. Those datasets are the widely known Wisconsin Breast Cancer dataset (Dua & Graff, 2017; Wolberg et al., 1995), Pima Indians Diabetes dataset (Smith et al., 1988), MNIST dataset (LeCun et al., 1998), and ImageNet-1k dataset (Russakovsky et al., 2015) (which was greyscaled). The different properties of these datasets are presented in Table 1.

We divided these datasets into a training set and a test set. For Wisconsin Breast Cancer and Pima Indians Diabetes we split the samples into 70% for the training set and 30% for the test set. In the case of the MNIST dataset, it comes with 10,000 samples for test. Thus, we took as training set the first 10,000 samples from the training dataset and the test set are those 10,000 test samples. Finally, for the ImageNet-1k dataset we took the first 10,000 samples of the 1,281,167 training samples for training and the 50,000 validation samples for test. The used Embodiments are the ones described in Section 3.1, with an overlap of 10% for Wisconsin Breast Cancer and of 50% for Pima Indians Diabetes due to their respective characteristics.

### 4.2 Experiments

The first experiment we performed aimed to test how well our proposal deals with a classification task compared to other unsupervised learning algorithms. Specifically, to K-Means with as many centroids as labels, and K-Means with the number of centroids that the elbow method (Thorndike, 1953) proposes. To evaluate the classification power of K-Means, each cluster was assigned the label that was most repeated between the training elements of that cluster. In the case of our proposal, the label selected is the one associated to the Pathway. Additionally, when comparing over MNIST, the Invariant Information Clustering (IIC) algorithm was computed too. In this case, the IIC algorithm was setup with the recommended parameters set by the authors for MNIST, and it was given the same number of epochs than our algorithm (1) and 100 epochs to give it some advantage. We could not try the author recommended number of epochs (3200) due to time and resource constraints.

To compare these algorithms, we executed them over the experimental subjects computing the learning curve. We display the resulting learning curves in Figure 3. Due to the size of the dataset, when computing the MNIST results we executed the experiments for the first 200 samples, from then on each 100 samples until the 2000 sample, and from then on each 1000 samples until the end.

The results of this experiment clearly show that our proposal is a better option for unsupervised classification. As we can observe, for tabular data our alternative is on par with K-Means for the Pima Indian Diabetes dataset (loosing by a 2.59%) and for the Wisconsin Breast Cancer dataset (loosing by a 1.76%). When we move to image data, we can observe how our proposal overcomes K-Means (as expected), but moreover, it is on par with IIC (winning by a 4.17%). During training there are points where IIC overcomes our proposal, as expected, but our proposal shows a quicker learning and wins in the final evaluation.

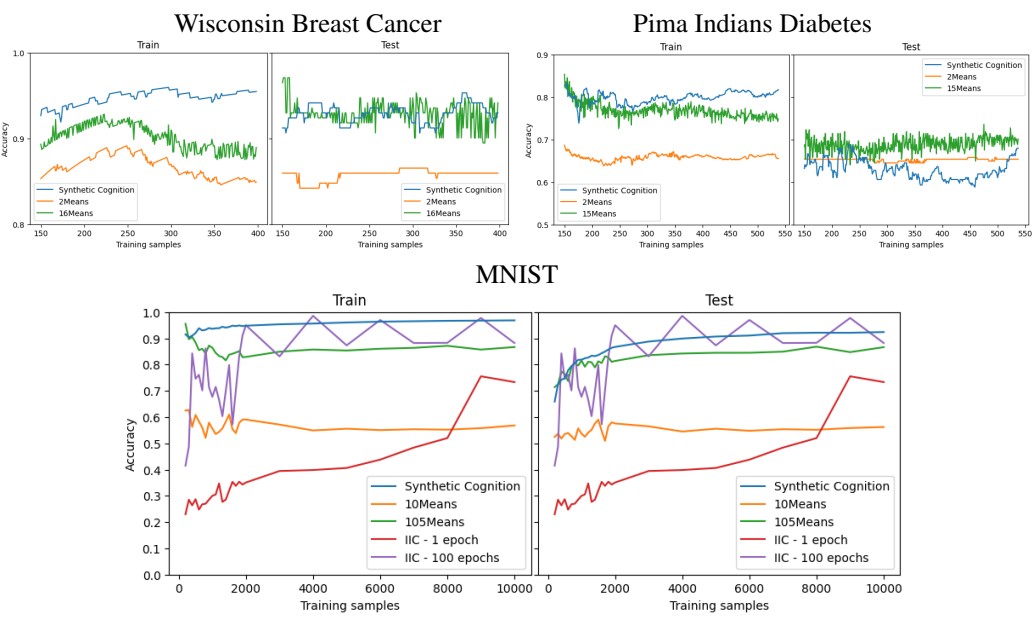

Figure 3: Learning curves comparison for the different datasets

We want to explicitly remark the fact that our proposal is able to obtain very good accuracies with fewer samples. For contrast, IIC needs around $1700$ training samples to obtain an stable accuracy over $70\%$ in test MNIST, while our proposal needs less than $300$ samples. Moreover, our proposal does not need multiple epochs to obtain such results: it only goes through the training samples once, although more epochs also improve results. If we compare with IIC with only $1$ training epoch, then IIC takes $9000$ samples to obtain that $70\%$ accuracy. Moreover, it is not able to overcome our proposal. If we give IIC an advantage of $100$ epochs, then IIC takes at least $2000$ samples to overcome our proposal for the first time. Moreover, after $10,000$ samples, our proposal beats the IIC-100 alternative, as it gets $92.37\%$ of accuracy compared to $88.2\%$ for IIC-100, what is more surprising considering that we executed our proposal with only one epoch.

Finally, our last experiment is in a more diverse and complex scenario: the unsupervised classification in the ImageNet-1k dataset. For this case, we transformed the images of ImageNet-1k into greyscale versions resized to size $256 \times 256$, and pre-processed with the Canny filter (Canny, 1986). In this case we decided to not compute the learning curve, in order to perform an stress test of our proposal. Thus, we ran our model over the dataset and obtained an accuracy of $0.092\%$. This result compares bad with respect to other algorithms, like the ones presented in Table 2. However, it is important to notice that those other algorithms were trained with the colored, non-scaled version of the dataset, and with $1,271,167$ more samples. More importantly, we did not use data augmentation techniques, or pre-training techniques, unlike the other algorithms presented in Table 2.

Additionally, we wanted to explore the cognition-like capabilities of our proposal, compared to other clustering algorithms, using noise distortion (Zhang et al., 2021). To that end, we devised an experiment using the MNIST dataset that consist on training the algorithms with the $10,000$ samples of the training set, and then take the $10,000$ samples from the test set and start taking out pixels. That is, for different percentages (from $0\%$ to $100\%$ with a step of $2\%$), we remove that percentage of pixels (that is, we set them to black) from all the samples of the test set. Then, we evaluate all the algorithms over that test set and compute both the accuracy curve and the area under such curve. We did the same experiment also using the $10,000$ train samples, in order to also evaluate such distortion curve over the already experienced samples. The selected clustering algorithms are: our proposal, our proposal capped to have only 1 cell, K-Means with 10 centroids, K-Means with 105 centroids, IIC with 1 epoch, IIC with 100 epochs, K-NN with 11 neighbours and K-NN with 1 neighbour. We display the resulting distortion curves in Figure 4.

| Image Classification Accuracy on ImageNet | |
| --- | --- |
| Method | Accuracy |
| Context (Doersch et al., 2015) | 30.4 |
| Color (Zhang et al., 2016) | 35.2 |
| Jigsaw (Noroozi & Favaro, 2016) | 38.1 |
| Contrastive (van den Oord et al., 2018) | 48.7 |
| Non-Parametric(Wu et al., 2018) | 54.0 |
| IIC(Ji et al., 2019) | 59.6 |
| Ours | XX.X |

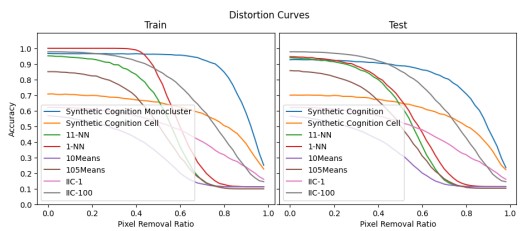

Table 2: Top-1 classification accuracy on ImageNet. Other methods values obtained from van den Oord et al. (2018) or their respective papers

Figure 4: Distortion curves comparison for the different clustering algorithms

The idea behind this experiment is that, even after removing some pixels from the image, humans are able to recognise numbers. Moreover, if given some specific pixels, and after being told that such pixels represent a number, humans are able to imagine and fill the number. Thus, we understand that recognising and/or reconstructing numbers is a capability of cognitive systems, one that we would desire in any Artificial Intelligence algorithm. Therefore, the goal here is to analyse how well each algorithm is able to recognise and reconstruct numbers from a set of pixels. As the pixels are removed at random, it is expected that after some removal percentage not even humans are able to recognise them, but the more pixels are removed, the better the concept of a number is understood if the correct number is recognised. Thus, this experiment is expected to set a difference between those algorithms that represent classes as subspaces and those that use constructive representations.

As we can clearly observe, our proposal has better distortion curves than any other alternative. In numbers, our proposal obtains an Area Under the Curve (AUC) of 86.92 over train and of 79.70 over test. When we limit our proposal to have only one cell, the AUC is of 59.30 for train and 58.92 for test. For K-Means with 10 centroids, the AUC is of 31.97 for train and 31.69 for test, and if we increase the number of centroids to 105 then the AUC is of 47.70 for train and 47.56 for test. Regarding IIC with 1 epoch, the AUC of both train and test is 47.07, and if we increase the number of epochs to 100 then the AUC of both train and test is 72.28. Finally, regarding K-NN, if we use 11 neighbours the AUC is of 55.36 for train and of 54.20 for test. If we reduce the number to be the closest neighbour, then the AUC is of 63.17 over training and of 56.75 over test.

The results also arise another interesting conclusion: the behaviour presented by our proposal is fundamentally different than the behaviour of the alternatives. The main difference is how well they develop when there is more than 50% of pixels removed. The compared alternatives have a step descend into a very low accuracy after this mark, showing that they have not properly modelled the subspace where those inputs with missing pixels fall. Thus, the label they assign to the input is selected almost randomly. In contrast, our proposal keeps getting high accuracy until the very end, when it is almost impossible to recognise the numbers even for a human (around an 80% of pixels removed). Moreover, this behaviour is still visible when limiting the algorithm to have only one cell. This clearly shows that our algorithm is doing something different than the other algorithms. We understand this is a direct effect of building constructive representations.

### 4.3 ABLATION STUDIES

In this Section we analyse the effect of the different parameters of our proposal. Let us start by stating that the main "parameter" of our proposal is the Embodiment. In this paper we have presented very basic Embodiments, and our algorithm works decently well with them. However, a fine tuned embodiment can cause huge increases in performance. For example, during our experiments with the Pima Indian Diabetes dataset we discovered that our initial embodiment (with a 10% overlap) did not obtain the best results. Thus, after testing multiple overlaps we settled in the 50% overlap. In general, for numerical data, the overlap between numbers is a fundamental parameter, because usually with a 0% overlap the performance is low, then it quickly raises with a small overlap and eventually falls down when the overlap is too big. Other "parameters" of our proposal are the similarity and Spatial Attention functions. We presented the ones that we have discovered that produce the best results,

after trying a lot of alternatives like the average of the similarity between Footprins for the Spatial Attention function or the Jaccard distance for the similarity function. Furthermore, we are aware that alternative functions can be developed with the potential to improve furthermore the results, but looking for those improved functions is matter of future work.

## 5 DISCUSSION AND LIMITATIONS

In this Section we would like to discuss the transparency and explainability of our algorithm, its capability of saying "I do not know", and its limitations.

Regarding transparency and explainability, it is fundamental to note that, as our algorithm has an internal hierarchical organisation of Sparse Distributed Representations (SDR), it is possible to recall how our algorithm decided which label corresponds to the input. To that effect, we need the decoder from the Embodiment to transform the internal SDRs to understandable outputs. Thus, we can interpret any decision as a filtering from the seed cell, based on its Footprints, and down the hierarchy until the last Footprint is activated. Then, composing the generated Projections, we build the Pathway, and the strongest label of the Pathway is the one selected.

Regarding the capability of our algorithm to say "I do not know", it is easily derived from our threshold setup. If a new input does not match with any Footprint in the seed cell, that is, its similarity with any Footprint from the seed cell is lower than the seed cell threshold, then our algorithm returns a value stating it does not know what is that input. This is in fact used during training to generate new Footprints. Moreover, that answer is not only an "I do not have a label for that input", but it actually means that it does not have a model for such input, so it can not return any Projection of it neither. This is an important and novel feature in an unsupervised learning algorithm. Its importance lies in the fact that saying "I do not know" ensures the user understands that the algorithm was not trained to recognise the pattern that was given, instead of falsely providing an answer and hallucinating (Ortega et al., 2021; Ji et al., 2023).

Finally, regarding the limitations of our proposal, its main one is the high memory costs involved compared to other alternatives due to the storage of a huge number of SDRs. We are aware that this limitation can hamper its scalability and applicability over very huge datasets and we are working in ways to diminish it, from developing growth inhibition and death mechanisms for the Footprints and cells, to improving our Embodiments to generate smaller SDRs.

## 6 CONCLUSIONS

Current well known unsupervised learning methods have a dim capability of extracting cognition-like relationships due to its assumption of a mathematical input space. The biggest exponent of this field is K-Means, that clusters samples based only on the mathematical distance between them. In this paper we have proposed an alternative representation-centric unsupervised learning algorithm to extract cognition-like relationships between samples through constructive representations.

Our proposal transforms the inputs into SDRs, and then generates an internal representation of those SDRs in order to later recall that representation when asked about the class of an specific input. We tested our proposal against K-Means and IIC for unsupervised classification tasks in four datasets, and show that our proposal is equivalent to them. Moreover, we have evaluated how well it can discover cognition-like relationships compared to other clustering algorithms, and we have found that it is better than the three main clustering algorithms: K-Means, IIC and K-NN. This is important because it means that our proposal does not only have a different, better behaviour than unsupervised learning algorithms, but also than supervised learning clustering ones.

As future work, we would like to explore how our proposal performs in other datasets and against other unsupervised learning algorithms, and perform an in deep analysis of the relevance of each "parameter" of our model. We would also like to develop new Embodiments for different input types, like colour images. We would like to explore the extension of our algorithm with other modulators too, like a conditioning modulator that allows us to have a reinforcement learning-like algorithm, or a temporal modulator that allows us to process sequences. Finally, we would like to explore different algorithms to compute the similarity function or the spatial attention function.

ACKNOWLEDGEMENTS

We want to thank [REDACTED]

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
