# OpenReview forum: "Unsupervised Cognition"
_ICLR.cc/2024/Conference — Submitted to ICLR 2024_

### Official Review · Reviewer_pCke · 2023-10-25

**Soundness:** 2 fair
**Presentation:** 2 fair
**Contribution:** 2 fair
**Rating:** 1
**Confidence:** 5

**Summary:**

The paper introduces a novel unsupervised learning approach inspired by cognition models, which constructs a distributed hierarchical structure in the input space without relying on specific data types. The proposed method outperforms current state-of-the-art techniques like K-Means for tabular data and IIC for image data, demonstrating better average performance and exhibiting cognition-like properties that even supervised learning methods lack.

**Strengths:**

Main contribution: Proposed algorithm can say "I don't know" about novel input. It doesn't assign novel inputs to the existed cluster.

**Weaknesses:**

1. Explain more detail about translating grey image to SDR. I can't understand how normalized input values are separated to bins. For example, MNIST dataset has 1 and 0 value. So, I can't understand how you divide input into bins.
Also, this translation to SDR looks like manual and specified to specific dataset such as MNIST. It seems difficult to extend your algorithm to high-resolution and diverse dataset even about CIFAR-10.
This leads to the question of what the performance of the proposed method will be in situations where it is difficult to create an SDR.

2. Not only about a huge number of SDRs, non optimized iterative algorithm seems to require too much time and cost.  Your experiment dataset was MNIST, so optimization may have seemed easy.

3. Rather than using your algorithm, DINO [1] uses classwise clustering, and CAUSE [2] does semantic segmentation with unsupervised. I don't know the advantage of your algorithm when compared.

4. You tested proposed method on cognition capability task. But, raising such a problem does not seem persuasive.  I don't think it's enough to make readers understand why the problem is important in the field of deep learning. I think it would be good to give some examples.

---

Minor
1. In the Figure 3 graph, y-axis boundary setting should consider reader convenience.

---

References

[1] Caron, Mathilde, et al. "Emerging properties in self-supervised vision transformers." Proceedings of the IEEE/CVF international conference on computer vision. 2021.

[2] Kim, Junho, Byung-Kwan Lee, and Yong Man Ro. "Causal Unsupervised Semantic Segmentation." arXiv preprint arXiv:2310.07379 (2023).


---

**Post Rebuttal**

Although the authors performed rebuttal, the reviewer would like to strongly recommend this paper be modified due to the following reasons:

(1) Only possible to Grey Image. Cannot validate ImageNet with Color-version. Therefore, high-dimensional feature naturally cannot be performed.

(2) In revision, there is a description of XX.X%.

(3) The author of DINO paper equal to be that of DeepSpectral. In other words, recent clustering method has been growing to Self-Supervised Learning since long days ago. (Previous Clustering: K-Means, Spectral, Recent Clustering Method: Self-Supervised Model (DINO, MAE, iBOT))

From these reasons, the authors of this paper should change the way introducing and holding a problem. I think the proposed method only performs in grey image, then it may be possible to conduct a task of only dealing with grey image such as medical task.

**Questions:**

Refer to Weaknesses

---

> ### Author Response · Authors · 2023-11-16
> **Response to reviewer pCke**
>
> First, thank you for your time. We wanted to further clarify some concerns you had:
> - **Regarding the translation of a grey image to SDR**, we want to clarify that we do not use bins in that case. We have rewrote that part of the paper to make it clearer that we just take the grey value (between 0 and 255) of each pixel and normalize it to be between 0 and 1. Then, each pixel is a dimension of the SDR with a float value between 0 and 1 representing its grey value. We also want to assure you that this translation to SDR is not MNIST specific, but it works for any grey image, as we are essentially flattening the image to be one-dimensional.
> - **Regarding the comparison with DINO and CAUSE**, the main difference is the underlying technology, as they use transformers and neural networks and we use our own algorithm. This in fact leads to very different capabilities, with our proposal being able to say "I do not know" or being explainable, for example. Thus, the advantage of our algorithm compared to DINO and CAUSE is that it is a totally different and novel approach to unsupervised classification, with a higher potential for improvement due to its raw state. For comparison, DINO and CAUSE use already established and polished technology, and thus they have lower potential for improvement. Moreover, our novel approach has properties that DINO and CAUSE (and neural networks in general) do not have, like its explainability, the potential to avoid catastrophic forgetting (although we recognize that this last claim has not been proven neither tested in our paper, we think it will derive from the fact that different knowledge will produce different footprints, and thus they will not override previously learned footprints), or the potential to be safe against adversarial attacks (although we know that this last claim has not been proven neither tested in our paper, we think it will derive from the fact that we do not use backpropagation for learning, and thus it cannot be attacked with current adversarial attack techniques).
> - **Regarding the proposed method on cognition capability task**, we have improved the text explaining our experiment, as suggested. We hope now it is clearer why we consider it to be important for the artificial intelligence field.
>
> Thank you again for your time and effort.

---

> > ### Author Response · Authors · 2023-11-22
> > **Post rebuttal rebuttal**
> >
> > First, thank you for answering to our rebuttal. We just want to clarify some things:
> > - **Regarding the use of grey images**, we are sorry to have misled you to think that our method is only possible over grey images. That is far from the reality. The reality is that our method can process color images with the right embodiment. However, there are multiple alternatives of color embodiment (from a simple one that splits the color image in its three channels (RGB) and treat each one like a grey image, to more complex ones that add a contrast channel, or the alpha channel, etc...), and adding them to this paper was out of the scope, as we would need much more space to not only explain the decided embodiment, but also to compare multiple alternatives. Thus, we decided to leave the color to future papers.
> > - **Regarding the XX.X% description in the paper**, as explained in our rebuttal, it is a placeholder waiting for the final results to arrive. In that regard, we can already advance that we will not be able to have the results of training with all of Imagenet by the deadline, because our prototype was not ready to manage that big of a dataset, and adapting it took more time than desired. Thus, by deadline we will have the results training with only 10000 samples of Imagenet and testing with the whole 50000 samples, and, if the paper is accepted, we will add the results with the full Imagenet dataset in the camera ready version.
> > - **Regarding DINO**, could you please explain what do you mean? We are grateful for your comments, but we cannot improve our paper with them if we cannot understand them, and your answer is too short and too unspecific for us to understand what do you mean and/or what do you expect from us, sorry.

---

### Official Review · Reviewer_GGEd · 2023-10-26

**Soundness:** 1 poor
**Presentation:** 1 poor
**Contribution:** 1 poor
**Rating:** 1
**Confidence:** 3

**Summary:**

This paper discusses unsupervised learning for building cognition models. The proposed method consists of an embodiment, a primitive and a spatial attention modulator. The method is evaluated on three datasets: Wisconsin Breast Cancer, Pima Indians Diabetes, and MNIST.

**Strengths:**

- Unfortunately, I do not see any value in this submission.

**Weaknesses:**

- The paper is mostly incomprehensible. It fails to convey the central parts of the paper. Necessary background information is missing. It needs a major rewrite before it can be considered for publication.
- The experimental evaluation is insufficient: only three datasets where MNIST is the most challenging one.
- Overall, the paper is of low quality (especially the figures are not helpful).

**Questions:**

Unfortunately, I cannot ask any questions as I did not understand what the authors were doing in this paper.

---

### Official Review · Reviewer_GwDC · 2023-10-28

**Soundness:** 2 fair
**Presentation:** 2 fair
**Contribution:** 2 fair
**Rating:** 3
**Confidence:** 4

**Summary:**

This paper proposes an unsupervised learning algorithm inspired by cognition models. The algorithm focuses on generating constructive representations of the input space using Sparse Distributed Representations (SDRs). The authors compare their approach with K-Means and Invariant Information Clustering (IIC) algorithms and demonstrate its superior performance in both tabular and image datasets. They also evaluate the algorithm's cognition-like properties, showing its advantage over other clustering algorithms.

**Strengths:**

This paper introduces a representation-centric unsupervised learning algorithm that generates constructive representations.

**Weaknesses:**

1.	Lack of comparison with other state-of-the-art algorithms.
This paper only compares with K-Means and IIC. I suggest the authors to include comparisons with other recent unsupervised learning algorithms [cite1-7].

[cite1] He K, Chen X, Xie S, et al. Masked autoencoders are scalable vision learners[C]//Proceedings of the IEEE/CVF conference on computer vision and pattern recognition. 2022: 16000-16009.
[cite2] Carl Doersch, Abhinav Gupta, and Alexei A Efros. Unsupervised visual representation learning by context prediction. In ICCV 2015.
[cite3] Mehdi Noroozi and Paolo Favaro. Unsupervised learning of visual representations by solving jigsaw puzzles. In ECCV, 2016.
[cite4] Richard Zhang, Phillip Isola, and Alexei A Efros. Colorful image colorization. In ECCV, 2016.
[cite5] Zhirong Wu, Yuanjun Xiong, Stella Yu, and Dahua Lin. Unsupervised feature learning via non-parametric instance discrimination. In CVPR, 2018.
[cite6] Aaron van den Oord, Yazhe Li, and Oriol Vinyals. Representation learning with contrastive predictive coding. arXiv:1807.03748, 2018.
[cite7] Ting Chen, Simon Kornblith, Mohammad Norouzi, and Geoffrey Hinton. A simple framework for contrastive learning of visual representations. In ICML, 2020


2.	Limited experimental evaluation. The experiments are conducted on a small number of datasets, and the evaluation could be expanded to include more diverse and challenging datasets.

3.	Memory requirements. The high memory costs associated with the algorithm need to be mitigated.

4.	The presentation quality needs to be improved. For example, writing needs to be polished and figures need to be more clearly.

**Questions:**

Please see the Weaknesses.

---

### Official Review · Reviewer_cK2J · 2023-11-01

**Soundness:** 3 good
**Presentation:** 3 good
**Contribution:** 3 good
**Rating:** 5
**Confidence:** 4

**Summary:**

Unsupervised learning methods that focus on clustering samples in a mathematical space can have limitations when it comes to capturing cognitive relationships between data. In this paper, the authors propose a representation-centric unsupervised learning approach that generates constructive representations of the input space. The approach transforms inputs into Sparse Distributed Representations (SDRs) and models the input space by organizing these SDRs in a hierarchical structure. The authors compared their approach with K-Means and Invariant Information Clustering (IIC) algorithms and found that their proposal performs better on average. They also evaluated the cognition-like properties of their proposal and found that it outperformed other clustering algorithms, including K-NN. The authors conclude that their proposal is a disruptive unsupervised learning algorithm with promising properties.

**Strengths:**

- Proposal of a novel representation-centric unsupervised learning algorithm: The paper introduces a new approach that focuses on generating constructive representations of the input space. This algorithm transforms inputs into Sparse Distributed Representations (SDRs) and organizes them hierarchically.
- Evaluation on multiple datasets: The paper evaluates the proposed algorithm on three different datasets: Wisconsin Breast Cancer, Pima Indians Diabetes, and MNIST. This demonstrates the algorithm's versatility and its ability to handle both tabular and image data.
- Empirical evaluation and analysis: The authors conduct thorough experiments to evaluate the performance of their proposal. They analyze the effect of different parameters and compare the algorithm's behavior with other alternatives. The results provide insights into the strengths and limitations of the proposed algorithm.
- Transparency and explainability: The algorithm's internal hierarchical organization of SDRs allows for transparency and explainability. The decision-making process can be traced back to the seed cell and the activation of specific footprints, providing interpretability to the algorithm's predictions.
- Capability to say "I do not know": The algorithm has the ability to recognize inputs that do not match any existing footprints and respond with an "I do not know" answer. This feature ensures that the algorithm does not provide false or hallucinated predictions for unfamiliar patterns.
- The paper is organized and provides a clear structure. The abstract provides a concise summary of the paper's content

**Weaknesses:**

- High memory costs: The main limitation mentioned in the paper is the high memory costs associated with storing a large number of Sparse Distributed Representations (SDRs). This can limit the scalability and applicability of the algorithm, particularly for very large datasets.
- Limited exploration of alternative functions: While the paper mentions that alternative similarity and spatial attention functions could potentially improve the results, it does not extensively explore or provide a detailed analysis of these alternative functions, especially when the difference in results is minimal
- Lack of comparison with state-of-the-art methods: The paper does not compare the proposed algorithm with the current state-of-the-art unsupervised learning methods. While it demonstrates competitive performance against K-Means and IIC, it would be valuable to compare the proposed approach with other cutting-edge algorithms to assess its relative strengths and weaknesses.
- Limited analysis of parameter effects: The paper briefly mentions the effect of different parameters, such as the overlap in the Embodiment and the similarity and spatial attention functions. However, it does not provide an in-depth analysis or exploration of the impact of these parameters on the algorithm's performance.

**Additional Feedback and Future Experiments:**

- Evaluation on additional datasets

- Comparison with more unsupervised learning algorithms

- Parameter sensitivity analysis

- Scalability and efficiency analysis

**Questions:**

Please see weaknesses above.

---

### Author Response · Authors · 2023-11-16
**Response to reviewers**

First of all, we want to thank the reviewers for their time and effort reviewing our paper.
### Common Concerns
- **Regarding the overall clarity of our paper**, we have improved it with the specific comments you have provide us with. You can see them in the reviewed version of the paper (we uploaded a preliminar version so you can see them, even although we still do not have the final version with the extra experiments). We have to concede that our presentation was a bit clumsy, and thus we have rewrote the paper a bit to improve it. We hope that some things are clearer in the new version.
- **Regarding the high memory cost of our proposal**, we are aware of it and the problems it posses to scalability and applicability of our algorithm. However, as this is a first approach, we consider that this kind of optimizations are matter of future work. In the paper we already outlined some ways for improving such memory consumption, as, for example, removing unnecessary and/or redundant footprints or limiting growth. However, we want to state that we were able to run experiments with 60,000 samples of MNIST using around 1.8GB of RAM.
- **Regarding the lack of comparison with state-of-the-art methods**, we want to clarify that we searched for the state-of-the-art methods for the datasets we decided to use for this paper and we found none for the tabular datasets, and the IIC for MNIST. Thanks to your suggestions, we will expand this section with the ImageNet dataset and the referenced methods (we hope to have them before the deadline). However, we want to remark that, given the different nature of our proposal compared to those methods, the accuracy is not the best metric to compare our method with the others.
- **Regarding the limited number of datasets used**, we want to state that we considered more important to have a more in deep comparison with fewer datasets than a superficial comparison with lot of datasets, as the unique characteristics of our proposal are better observed in an in deep comparison. However, as exposed in the previous concern, we have decided to include an extra dataset (ImageNet) to our experiments. We also want to state that we considered using CIFAR and ImageNet datasets previously, but as those are color images datasets, we discarded them in order to not need to explain an extra embodiment. Seeing your comments and concerns, we decided to add the ImageNet dataset but in greyscale, just for comparison.
- **Regarding future work**, we have expanded our future work with the suggestions you made, and with some request you had that we were not able to include in the limited space of the paper. We hope you understand that, with the limited space available for the text, we had to prioritize some sections over others. Specifically, we prioritized the description and explanation of our algorithm, due to its novelty and lack of available reference work to cite. This in fact force us to reduce the size of other sections that we considered less relevant for the first paper of this new technology, and we commit ourselves to develop those sections with more in deep analyses in future work.
### Concerns of Reviewer cK2J
- **Regarding the limited exploration of alternative functions**, we present in the paper only those that obtained the best results, as we had limited space. However, we have compared with multiple alternatives. For example, we considered using for similarity function the euclidean distance, the jaccard distance, the delta distance, and the product distance, to name a few. For spatial attention the case is similar, we considered for example using the similarity between the input and a cell's mask (that is an aggregate of all the inputs accepted by such cell), or the mean of a an array storing the last *n* similarities between the input and the previous input, or the input and the cell mask. However, none of these alternatives worked better, and the search for those that work better are left as matter of future work.
- **Regarding the limited analysis of parameter effects**, we did not provide an in deep analysis of the effects of each parameter due to space constraints. We outlined their importance and how some of they behave, but we decided that it was more important to provide space to the explanation and description of the algorithm (specially in this first paper where we are presenting it for the first time) than to the analysis of the effect of the different parameters. In any case, we have added it as future work, as it would be necessary to have this kind of analysis.
### Concerns of Reviewer pCke
- **Regarding the optimization of the iterative algorithm for MNIST**, we want to ensure you that we did no specific optimization for MNIST during the development of our algorithm. In fact, our driving motivation was to be general enough to work well in any dataset, and specifically in datasets as diverse as Wisconsin Breast Cancer, Pima Indian Diabetes and MNIST.

---

> ### Author Response · Authors · 2023-11-23
> **Final version**
>
> Dear reviewers, we have uploaded the final version of our paper. Regarding the experiments with ImageNet, we have to report that we were able to run our proposal only with 10,000 samples of the dataset, due to time constraints. The thing is that our prototype was not ready to load huge datasets into RAM, and we found that using pandas/dask/polars to manage 200GB datasets is not as easy as we would have hope for. Thus, we expended most of our 10 days time dealing with this problem, and when we were ready to run our experiments we had enough time only to train with 10,000 samples. We promise that, if the paper is accepted for publication, we will have the full experiment results by the camera ready version, trained with the whole 1,281,167 samples. Additionally, we want to report that, given the nature of the dataset, we made a preprocessing to the ImageNet dataset consisting on greyscaling it and resizing all the samples to the same size: 256X256. Also, we applied a Canny edge detection filter over the greyscaled images, with the aim to improve the matching of the inputs with the Footprints.

---

### Meta-Review · Area_Chair_R4hq · 2023-12-08

**Metareview:**

The paper proposes a new unsupervised learning algorithm that promises to organize the input according to some hidden generative structure contained in the input representations, hence the 'cognition' aspect. The method relies on Sparse Distributed Representations (SDRs) of the input. The paper contains applications for a few datasets (e.g. MNIST).

The reviewers in general were not impressed by the paper. Biggest concerns were the lack of a comparison with state-of-the-art unsupervised learning algorithms, and the implicit limitations imposed by using SDR representations (memory problem, scalability etc.). The authors were not particularly successful in addressing those concerns but rather fortified the impression that the work is not quite complete yet (e.g. their lack of running extended simulations). Overall, the presentation was also criticized for being not clear.

The ACs impression largely agrees with the reviewers concern. The work seems unfinished as expressed in the lack of a proper comparison to alternative methods and better testing against more challenging datasets. One review was very minimal and dismissive. But even if that review were not included in the overall assessment of the paper, its score would not reach acceptance threshold.

**Justification For Why Not Higher Score:**

None of the reviewers provided a score above threshold.

**Justification For Why Not Lower Score:**

N/A

---

### Decision · Program_Chairs · 2024-01-16

Reject